# Non-Invasive Probing of Winter Dormancy via Time-Frequency Analysis of Induced Chlorophyll Fluorescence in Deciduous Plants as Exemplified by Apple (*Malus* × *domestica* Borkh.)

**DOI:** 10.3390/plants11212811

**Published:** 2022-10-22

**Authors:** Boris Shurygin, Ivan Konyukhov, Sergei Khruschev, Alexei Solovchenko

**Affiliations:** 1Faculty of Biology, Lomonosov Moscow State University, Leninskie Gory 1/12, 119234 Moscow, Russia; 2Institute of Natural Sciences, Derzhavin Tambov State University, 392036 Tambov, Russia; 3Federal Scientific Agroengineering Center VIM, 109428 Moscow, Russia

**Keywords:** chilling requirement, chlorophyll fluorescence, non-photochemical quenching, PAM, photoprotection, stress resilience, winter dormancy

## Abstract

Dormancy is a physiological state that confers winter hardiness to and orchestrates phenological phase progression in temperate perennial plants. Weather fluctuations caused by climate change increasingly disturb dormancy onset and release in plants including tree crops, causing aberrant growth, flowering and fruiting. Research in this field suffers from the lack of affordable non-invasive methods for online dormancy monitoring. We propose an automatic framework for low-cost, long-term, scalable dormancy studies in deciduous plants. It is based on continuous sensing of the photosynthetic activity of shoots via pulse-amplitude-modulated chlorophyll fluorescence sensors connected remotely to a data processing system. The resulting high-resolution time series of JIP-test parameters indicative of the responsiveness of the photosynthetic apparatus to environmental stimuli were subjected to frequency-domain analysis. The proposed approach overcomes the variance coming from diurnal changes of insolation and provides hints on the depth of dormancy. Our approach was validated over three seasons in an apple (*Malus × domestica* Borkh.) orchard by collating the non-invasive estimations with the results of traditional methods (growing of the cuttings obtained from the trees at different phases of dormancy) and the output of chilling requirement models. We discuss the advantages of the proposed monitoring framework such as prompt detection of frost damage along with its potential limitations.

## 1. Introduction

Winter dormancy is a physiological state characterized by physiological quiescence and hence by increased resilience to harsh environmental conditions such as low temperatures. It is among the key adaptations of perennial plant species, including temperate annual and deciduous plants, for survival during the cold season [1].

Winter dormancy proceeds through the three phases: the first phase is pre-dormancy, during which mechanisms increasing stress tolerance of plants are deployed [2]. It is followed by physiological or deep dormancy, endo-dormancy [3,4]. Endodormancy is defined as inactive state of meristems and/or organs capable of growth in which growth does not resume even under favorable conditions until this state is changed by environmental signals [5,6]. The onset and release of dormancy is synchronized with seasonal climate changes; as noted above, it ensures preservation of shoot tissues as well as vegetative and generative buds during the cold season [7,8].

Upon exposure to low temperatures during the cold season, plants progress from endo-dormancy to the next phase—eco-dormancy—when growth is restricted only by unfavorable climatic conditions; this event is called dormancy release [2,4]. Although the mechanisms of the dormancy onset, maintenance and release are still being debated, these phenomena are believed to be jointly controlled by the expression of certain genes, hormonal balance and environmental stimuli such as temperature and photoperiod [9,10,11,12]. Thus, the release of endo-dormancy requires exposure to low temperatures termed chilling requirement (CR), which is expressed in chilling units (CUs) and reflects the amount of plant exposure at low temperatures, most commonly in the range of 0 to 7 °C. The CR varies depending on genotype and regional climate [13,14]; it can be predicted using several mathematical models (see [15] and references therein). Effects of insufficient exposure to low temperatures have already been reported for many commercially significant cultivars—peach (*Prunus persica*) [16,17], pear (*Pyrus* L.) [18], plum (*Prunus* sect. *Prunus*) [19] and apple (*Malus × domestica* Borkh.) [20], among others; see also [21,22] and references therein. Most commercial cultivars of apple are especially vulnerable to these effects, exhibiting a relatively high chilling requirement ranging typically between 900 and 1500 CUs depending on the model used [23]; walnut, pear and plum have high chilling requirements as well [24].

Increasing research interest in winter dormancy is also fueled by its practical importance for crop production, especially for growing fruits in regions with unstable climatic conditions. The problem of insufficient chilling exposure has become especially urgent in recent years [8,22]. Weather fluctuations, causing aberrant dormancy onset or its premature release, impair stress resilience of plants, making them vulnerable to frost snaps common in the beginning of the vegetation season and deteriorating floral bud formation and fruit set. These effects are already being observed in warm winter regions [22,25]. Therefore, researchers and breeders require estimations of dormancy depth for plant phenotyping and breeding for hardiness. Determination of the CR and corresponding growing zones for a given cultivar are high priority tasks for agricultural sustainability and food security, but they are challenging to perform at scale because of the destructive nature of the dormancy assessment. Despite the importance of the problem, only a limited set of methods is available for studies of dormancy. Both conventional (growing of the cuttings taken from plants at different phases of dormancy) and recently developed methods (e.g., those gauging the expression of dormancy-related genes) [11,26] are invasive. To overcome this limitation, a technique for online assessment of dormancy state is desirable.

A plausible approach compatible with these requirements is based on recording and analyzing the amplitude-kinetic characteristics of pulse-amplitude-modulated (PAM) induced chlorophyll fluorescence (CF) widely used in high-performance plant phenotyping [27,28,29]. In this approach, chlorophyll *a* serves as an “internal probe” of the photosynthetic apparatus (PSA) condition reflecting indirectly the metabolic activity of tissues; hence, it is expected to reveal inter alia the dormancy status of plants.

The CF/PAM-based techniques became widespread in the past decades as a promising approach to vegetation sensing, giving deep insights into various aspects of the physiological condition of plants and their stress resilience [30,31,32]. Thus, a CF/PAM-based protocol was developed for assessment of winter hardiness [33] and dormancy of conifers [34], but it did not allow for separating the effects of low temperature and those of solar radiation intensity. In deciduous plants, including tree crops, CF induction curves can be recorded directly from the shoots with phelloderm containing chloroplasts capable of photosynthesis [35,36,37,38], such that the CF/PAM-based approach was applicable to deciduous plants throughout the cold season [33], but it has not been automated or carried out with time-resolved measurements so far. 

In *Fagus sylvatica*, leaves and shoots demonstrated the same pattern of Fv/Fm changes in the spring–summer period [38]. It was also shown that CF of the shoots was related to the cold resistance of willow clones and the degree of frost damage to their cambium [39]. Current evidence suggests that the temperature dependence of the CF parameters can be, in principle, used as a proxy for identifying cold-resistant plants. Still, CF-based approaches capable of resolving the onset and release of winter dormancy (endo-dormancy) are so far lacking. Winter hardiness of dormant conifer plants was associated with the induction of a high, slowly relaxing NPQ dependent on phosphorylation of thylakoid proteins and upregulation of the violaxanthin cycle [40,41,42]. Another luminescent marker of dormancy is thermoluminescence: its B-band declines at early stages of pine needle hardening, likely due to increased charge recombination in PS II providing additional photoprotection [43]. Taken together, the published reports available to the authors of this work do not present a conclusive approach to non-invasive assessment of winter dormancy via CF induction curves, especially that suitable for deciduous trees. At the same time, new and upcoming climate challenges result in an increased demand for an affordable, scalable and non-invasive approach to dormancy monitoring from researchers, breeders and growers alike.

Here, we report on a novel approach to record and treat CF data for non-invasive online assessment of dormancy depth of and frost damage to trees exemplified by apple. It assumes that physiological and biochemical rearrangements accompanying the onset and release of dormancy in plants also trigger measurable changes in the functioning of the photosynthetic apparatus (PSA). In turn, these changes are reflected in the long-term dynamic of the CF/PAM parameters known as the JIP test [30,44,45,46], which are further subjected to time-frequency analysis.

## 2. Results and Discussion

To find an approach to selective and sensitive non-invasive monitoring of the onset and release of winter dormancy in the model deciduous plants, a long record of CF/PAM and weather data obtained across seasons over several years was studied. Resultant periods of conjectured endo- and eco-dormancy were validated against traditional criteria of dormancy, and the corresponding arrays of non-invasively recorded data were subjected to multi-step mathematical analysis to reveal parameters potentially indicative of the dormancy depth.

### 2.1. Winter Dormancy and Dormancy Release

According to multi-year observations, in the growing region where our experimental plot is situated, the onset of winter dormancy in apple trees normally starts at September–October and completes by mid-November. The release of endo-dormancy and transition to eco-dormancy occur, depending on the specific season conditions, from the end of January until mid-February, while the budbreak is observed in the end of April, and the bloom takes place in the middle of May.

To establish the objective reference for the non-invasively obtained data, we tested the depth of dormancy using a traditional destructive test—growing of the cut-off shoots under room conditions. Since it was not possible to cut shoots frequently from the experimental trees due to their young age, the destructive tests were used only to validate the key phases of dormancy (eco-dormancy and endo-dormancy) in each season. Under our experimental conditions, the shoots collected in late November remained dormant (little or no budbreak was observed, Table 1). In contrast, the shoots sampled in the middle of February displayed budbreak and subsequent growth of leaves and, in many cases, developed flowers within 10–14 days. 

Overall, the results of the destructive tests carried out in this work were in line with the multi-year observation on the onset and release of winter dormancy in apple trees in the location where the experimental orchard was planted. An exception was observed in the season 2020 featuring a warm winter (average temperature in November–December was above −5 °C), when the dormancy progression was highly erratic. The time points corresponding to the sampling of shoots displaying the absence or presence of budbreak under favorable conditions are marked on the figures below. In the field, the budbreak in the trees of the experimental orchard plot was observed in the third decade of April.

### 2.2. Chlorophyll Fluorescence Transients

The work was started by pilot experiments on comparison of the CF transients (commonly designated as OJIP curves, see also Appendix A) taken from the shoots incubated at different temperatures: −18 °C (a common winter temperature at the site of the experimental orchard), warmed up to 25 °C and frozen at liquid nitrogen temperature (around −195 °C). The OJIP curves recorded at −18 and 25 °C (Figure 1, curves 1 and 2) were similar and contained the features characteristic of the PSA of healthy plants (for more details, see, e.g., [45]). At the same time, the OJIP curves of the shoots kept at –18 °C were characterized by a lower overall amplitude resulting in a lower Fv/Fm. 

The OJIP curves of the shoots incubated in liquid nitrogen (LN) were dramatically different in their shape from those taken from the intact shoots (*cf.* curves 1, 2 and 3, 4 in Figure 1). Notably, the shape of the LN-frozen curves did not change significantly after thawing the LN-frozen shoots. Obviously, the observed changes reflect the destruction of the PSA and, ultimately, death of the shoot tissues. The latter conclusion was supported by the absence of budbreak and growth of the LN-frozen shoots further incubated at 25 °C.

Overall, the healthy cold-acclimated apple tree shoots exhibited measurable CF transients with characteristic features of viable plant tissues. On the contrary, irreversible frost damage has changed the shape of the chlorophyll fluorescence transient dramatically. Accordingly, our pilot measurements showed the feasibility of the recording of OJIP curves from sound apple shoots at cold season. As an added benefit, analysis of the OJIP curves was capable of rapid online detection of frost damage without the need for laborious laboratory tests and observations.

### 2.3. Long-Term Correlations of the CF Parameters and Weather Conditions

To monitor the CF transients directly from the shoots of the trees planted in the orchard, the custom-made PAM sensors (Appendix A) were made capable of wireless transmission of hourly measured OJIP curves to a remote server for processing and visualization and the examples of raw CF transients (Appendix A). In total, over 35,000 transients and over 16,000 weather measurements were collected and processed over the observation period. To establish candidate parameters for further analysis, a set of correlation matrices was produced for the complete dataset (Figure 2) as well as for the data obtained around midday (i.e., with the highest sunlight intensity, Appendix A) or midnight (Appendix A). The number of observations processed ensured that all of the correlation values that are not close to zero are statistically significant, with *p*-values being below 1 × ^−5^ and, more typically, in the order of 1 × ^−200^.

Of all meteorological parameters monitored, only air temperature strongly correlated with the condition of PSII (e.g., r [Fv/Fm vs. air t°] = 0.75). Certain JIP-test parameters included in the initial dataset were, by definition, inter-correlated and hence redundant. These parameters are not shown in the figure but can be found in plots in Appendix A, as well as plots of Spearman’s rank correlation, which were found to differ insignificantly from Pearson’s r-values. Still, the analysis of correlation coefficients calculated for the complete dataset across seasons and years (designated as “bulk correlations” here) yielded several useful hints. Thus, maximal quantum yield of photosystem II (PS II), Fv/Fm (see Appendix A), the parameter by far the most frequently used as an integral indicator of physiological condition of plant organism [47], showed a strong correlation with air temperature (Appendix A) but not with other weather parameters, indicating a sizeable influence of this factor on the performance of the PSA. 

Surprisingly, the correlation of the studied parameters with the flux of thermally dissipated energy (DI_0_/RC) was low (Figure 2). All JIP-test parameters studied demonstrated a weak correlation with UV radiation intensity and projected total solar radiation fluxes. Collectively, the results of the “bulk correlation” analysis suggested that the air temperature exerted a more profound effect on the condition of PSA of the trees during their winter dormancy as compared with light intensity. At the same time, the *r*-values for these correlations were generally low.

Temperature correlation: Since the selectivity and sensitivity of the “bulk correlation” analysis was rather limited, we more closely observed monthly average values of the JIP-test parameters. Most of the parameters analyzed displayed a high variation and a lack of conclusive trend (not shown). A prominent exception was the Fv/Fm and DI_0_/RC parameters (Figure 3). Thus, dark-adapted (measured at night) values of Fv/Fm tended to decline during the onset of endo-dormancy and increased when the dormancy was released; light-adapted values demonstrated an opposite trend (Figure 3a).

In the case of DI_0_/RC measured under daylight conditions (Figure 3b), there was a pronounced decline during the onset of endo-dormancy followed by a sharp increase in this parameter when the dormancy was released. Notably, the magnitude of changes in DI_0_/RC was much higher than in Fv/Fm (3-fold vs. around 1.5-fold, respectively). Nighttime values of this parameter, as expected, did not show a clearly visible trend of changes.

In view of the findings outlined above, we hypothesized that the depth of winter dormancy in the studied plant object is somehow related to responsiveness of the photoprotective responses to the PSA to harsh environmental conditions. Accordingly, Fv/Fm and DI_0_/RC were selected as candidate parameters for further analysis on the grounds of their high dynamic range and relatively low noise. On one hand, the engagement of the photoprotective mechanisms depends on the incident PAR flux and the presence of other stresses such as low air temperature. On the other hand, we did not see a pronounced difference in the correlations between the parameters measured at daylight and at night. Therefore, we attempted to deconvolute the time-frequency behavior of the kinetics of Fv/Fm and DI_0_/RC using the time-resolved data recorded hourly across seasons (see Materials and Methods).

### 2.4. Time-Frequency Analysis of the CF Data

For purposes of forward analysis, an attempt was made to predict Fv/Fm solely from recorded temperatures and insolation estimates (see Materials and Methods, Equation (2)). While the resulting approximation displays a significant amount of inter-dependency between parameters, such a simple model could not capture complex relations and over-estimated the effects of temperature (Appendix A), yet at r^2^ = 0.68 (RMSE = 0.12), the goodness of fit is remarkably high as compared to that for DI_0_/RC (at just r^2^ = 0.07; RMSE = 191.4)—the fraction of explained variance interpretation thus suggests that Fv/Fm is highly volatile. That is, Fv/Fm appears to be affected to a high degree by the transient, high-frequency environmental changes—specifically, changes in insolation and temperature. Conversely, DI_0_/RC remains largely unaffected by those outside of eco-dormancy, which is further corroborated by the analysis below.

Autocorrelation plots (Appendix A) have demonstrated remarkably strong daily periodicity of Fv/Fm but vanishingly small day-to-day consistency for parameters reflecting fluxes of energy per reaction center such as DI_0_/RC, ET_0_/RC and TR_0_/RC. In *P. silvestris* and *Picea abies*, despite significant differences in their mechanisms of acclimation to low temperature, Fv/Fm follows the time-course of ambient temperature [49]. In *P. silvestris*, a decline in Fv/Fm can be used to predict cold resistance [50]. Thus, for late-flowering almond (*Prunus dulcis*) varieties with pronounced susceptibility to frost, a linear decrease in Fv/Fm with temperature was found, and for early-flowering varieties of the same species resistant to low temperatures, a quadratic curve with an inflection point at −1 °C was observed [51]. However, the actual (operational) quantum output of PS II is more sensitive than Fv/Fm, because it can change rapidly [52], while a decrease in Fv/Fm can be detected only at the deep stages of hardening or winter stress [53].

Under our experimental conditions, the parameters expressed on an RC basis were more informative with respect to the transition throughout phenological stages and stable downregulation of PSII activity. Indeed, earlier research highlighted the strong connection between chilling and photoinhibition, which is especially pronounced in apple trees (see [54] and references therein).

Lowpass filtering was found to produce relatively smooth resulting curves starting from the filter transition width corresponding to one week (Figure 4). The frequency response of the filter and results for other JIP-test parameters can be found in Appendix A. After filtration, these curves reveal that Fv/Fm per se is a sensitive but not sufficiently selective indicator of cold acclimation and dormancy, which are known to go hand in hand in apple [55]. Thus, during the unusually warm winter of 2019/2020, Fv/Fm hardly dipped below the values typical of the vegetation season.

The DI_0_/RC parameter exhibits a similar trend of change, but the fine structure of its broad winter peak is much more pronounced and easier to study numerically (Figure 5). Of special interest is the period corresponding to the winter of 2020–2021, which was mild with average temperatures around −1 °C. Under such conditions, the increase in DI_0_/RC is barely detectable as compared to frosty winters of two subsequent seasons (Figure 5, see also Figure 6b below). The dormancy release in the season 2020–2021 was also delayed by at least three weeks.

Wavelet analysis carried out as described in Materials and Methods (see also the Appendix A) confirmed the hypothesis that DI_0_/RC conveys more information about the dormancy depth in apple than Fv/Fm. The Fv/Fm spectrogram shows (Figure 6a) a separation between daily cycles (top horizontal black line, bottom black line corresponds to a period of one week) and slower changes owing principally to weather conditions. The general trend of changes in spectral power (which is assumed to reflect the magnitude of relative contribution of the processes and/or stimuli with corresponding frequency of oscillation) is hard to uncover, and the spectrogram is noisy overall, indicating many processes affecting Fv/Fm are happening at once. While Fv/Fm could be used to trace general trends in data, it is not selective enough to describe changes specific to certain phase of dormancy.

In contrast, the DI_0_/RC spectrogram (Figure 6b) shows clear peaks corresponding to eco-dormancy phases; a similar pattern is evident on the plots with the results of lowpass filtration of the DI_0_/RC signal (Figure 5). Importantly, the prominent peaks of spectral power evident on both types of plots corresponded to the periods of time when dormancy was already released and eco-dormancy took place, according to the destructive tests with the cut shoots.

Although both Fv/Fm and DI_0_/RC reveal valuable information about the physiological condition of the plant, all of the analysis performed supports the conclusion that the dormancy in woody plants is better described in terms of the oscillation of photoprotective thermal dissipation in response to the cold and high sunlight stresses rather than by quantum yield of PS II. DI_0_/RC also holds promise as a measurable marker of the mechanisms driving the metabolic changes during dormancy, whereas Fv/Fm is an integral characteristic not specific to any single type of stress. While Fv/Fm is better suited for the overall assessment of the PSII state, we suggest the kinetics DI_0_/RC and other per-RC characteristics as candidates for the research on cyclic processes with relatively long periods (on the scale of weeks and months).

### 2.5. Agreement of the Non-Invasive Dormancy Assessments with Those Derived from the Common CR Mathematical Models

To further validate our conclusions regarding the applicability of the proposed approach, we compared our assessments of dormancy status of the experimental plants with predictions yielded by the most widespread mathematical models [15]: the model based on chilling hours [56], the Utah model [57], the “low chill” model [58] and the North Carolina model [59] from the weather information collected during the observation period. The predictions of the periods when eco-dormancy release was expected were in reasonable agreement with the assessments made on the basis of the time series analysis (Figure 7), further supporting the validity of the approach developed here.

## 3. Materials and Methods

### 3.1. Plant Material and Experiment Design

The experiments were carried out at an experimental orchard (0.1 ha) located in the Botanical Garden of Lomonosov Moscow State University (55.7078 °N, 37.5268 °E) using three-year old apple (*Malus × domestica* Borkh.) var. “Flagman” plants grafted on B-396 rootstock planted according to the 2.5 × 0.8 scheme and trained as “spindle” (see also Appendix A). The planting density was 4000 plants ha^−1^. The experiment was carried out throughout October 2019–May 2022. 

To confirm the onset of endo-dormancy indicated by the online non-invasive measurements (see below), five 20 cm shoots were cut from five experimental trees (for the details on placement of the experimental trees within the field site, see Appendix A) in the end of November–beginning of December (see Table 1) and grown in tap water under room conditions (+22 ± 25 °C, 45–55% relative humidity, ambient light and photoperiod). To confirm the release of endo-dormancy, the same number of shots was cut in the end of January–mid-February (see Table 1) and grown under the same conditions. The absence of budbreak within a 10-day period was considered a confirmation of the onset of endodormancy, whereas budbreak, growth and development of leaflets and flowers (Figure 8) were accepted as a confirmation of dormancy release. 

The frost damage simulation experiments were conducted after release of endo-dormancy (February 2019). The shoots were cut at the orchard at 10:00 a.m. and kept at −18 °C in darkness. For the five control samples, OJIP curves were recorded (i) at −18 °C and (ii) after warming them up to +25 °C in darkness. For the other five shoots, OJIP curves were recorded at −18 °C, then the shoots were dark-frozen in liquid nitrogen (−195 °C) for 30 min. The OJIP curves were then recorded from the frozen shoots after warming them up to +25 °C, also in darkness.

### 3.2. Chlorophyll Fluorescence and Weather Data Collection

For recording Chl fluorescence induction curves of the shoots incubated indoors, FluorPen S100 (Photon Systems Instruments, Drasov, Czech Republic) and its built-in “OJIP” protocol was used. For continuous outdoor measurements, two in-house made PAM-fluorometers were fixed on the internodes of 2-year shoots (around 1.5 cm in diameter) in the middle part (ca. 1.5 m above ground) of the canopy of two different apple trees (Figure 9a,b). The PAM sensors employed the fast-repetition-rate technology, FRR [60], and possessed the same characteristics as the earlier developed devices [61]; for more detailed information, see the description of the device in the Appendix A.

The experiment was started with a single PAM fluorometer (19 October 2019–6 February 2020). After that, a second unit was added for redundancy and cross-validation. Once an hour, each PAM sensor transmitted the measured fluorescence transients (OJIP-curves) to an in-house developed remote server for archiving, processing (see below) and visualization (Figure 9c). An automatic weather station (Sokol-M, GSK Escort, Kazan, Russia) was remotely connected to the same server to complement the CF/PAM data with current weather parameters (air temperature, relative humidity, atmospheric pressure, precipitation, wind strength and direction, as well as solar UV intensity).

### 3.3. Analysis of Fluorescence Transients (JIP Test)

Raw fluorescence transients were routinely processed using the pyPhotoSyn software [62] to calculate JIP-test parameters according to Strasser et al. [48]. Minimal fluorescence F_0_ and initial slope of the fluorescence transient M_0_ were estimated as the intercept and the slope of linear approximation of the initial linear segment of the fluorescence transient according to Plyusnina et al. [63]. For the complete list of JIP-test parameters, see Appendix A.

### 3.4. Data Processing and Analysis

Further processing was carried out in Python (version 3.8). First, the weather data were interpolated to the grid defined by the PAM fluorometers timestamps. Parameters derived from the JIP test were cleaned up, with values of far-flung outliers set to zero at the first stage of processing. Negative values and values exceeding the normal range more than tenfold were filtered out with this method.

After that, exploratory data analysis was performed: JIP-test parameters were subjected to cross-correlation and auto-correlation analysis. To estimate the apparent dependence of the parameters on transient environmental conditions, multiple linear regression was used. This qualitative reconstruction of the parameter dynamics from insolation and temperature was then considered an indirect measure of how much information a given parameter provides about short-term vs. long-term plant adaptation. Finally, time-frequency analysis of the PSII characteristics as revealed by JIP-test was performed.

For modeling purposes, the entire dataset was augmented by solar elevation angles with the help of skyfield package using DE431 ephemeris [64]; these angles follow directly from the location, date and time of the measurement and provide an estimate of insolation. Potentially available sunlight was approximated by the formula
I_D_ = 1.353 × 0.7^AM^0.678^^, AM = 1/cos(θ),(1)
where AM is air mass, θ is the solar zenith angle (complementary to the solar elevation angle) and I_D_ is the direct component of insolation (in kW m^−2^) [65]. The parameters yielded by the JIP test were then fitted by a multi-linear function of temperature and direct sunlight:X = k_1_ + k_2_ × T + k_3_ × I,(2)
where T is temperature (in degrees Celsius) and I is insolation (in kW∙m^−2^). This model is intentionally simplistic and does not take into account cumulative effects of exposure to low temperatures and sunlight; it allows for the goodness of fit to be used as a metric. Here, the higher *r*-squared for modeling a given parameter, the more this parameter is affected by transient changes and the less it could reveal about the long-term dormancy-related changes.

Time-frequency analysis was performed on a subset of parameters chosen using correlation analysis (both Pearson’s *r* and Spearman’s *ρ* were calculated to capture both linear and non-linear relationships) and multi-linear regression models described above. Autocorrelation curves were also produced for estimating periodicity in signals as an indirect measure of dependence of a given parameter on changes in insolation and temperature driven by daily cycles. Spectrograms were produced using the pycwt package using a Morlet mother wavelet with the default value of w_0_ = 6 and frequencies corresponding to the range of 1 to 512 h. Lowpass filtering using Fourier transform, and the cutoff frequency of (one week)^−1^ was also employed.

Relevant code used in the analysis is available on GitHub (https://github.com/Lodinn/PAM-timeseries, accessed on 20 September 2022)).

To infer predictions of the CR accumulation sufficient for dormancy release, we employed the most widespread mathematical models [15]: the “chilling hours” model (Weinberger, 1950 [56]), the “Utah” model (Richardson et al., 1974 [57]), the “low chill” model (Gilreath and Buchanan, 1981 [58]) and the “North Carolina” model (Shaltout and Unrath, 1983 [59]).

## 4. Concluding Remarks

Dormancy is a complex phenomenon controlled by superposition of environmental stimuli (chiefly by photoperiod and temperature), mechanisms of perception and transmission of these signals, as well as the plant’s responses to them. Our understanding of the mechanisms of the induction and regulation of winter dormancy and its phenotypic manifestations at the level of PSA is clearly insufficient. To overcome this limitation, we attempted to develop an approach to non-invasive express assessment of the winter dormancy depth in deciduous plants via CF induction curves using apple trees as a model. This problem has been tackled previously [33,50], but the previous attempts assessed, directly or indirectly, a magnitude of photoprotective response of PSA to the combined action of low air temperature and sunlight. Although CF is a sensitive probe of plant condition, it is easily affected, e.g., by variations in ambient illumination conditions. Furthermore, acclimation to diverse factors such as cold and light intensity converges on PSA functioning [66,67] interfering with the manifestation of intrinsic processes such as the onset release of endo-dormancy. Indeed, a significant complication of the non-invasive approaches is the need to disentangle the effects of diverse environmental stimuli and intrinsic responses connected by a complex web of regulatory mechanisms. Long-term (seasonal) trends, e.g., those associated with dormancy, are obscured by short-term responses to weather fluctuations profoundly affecting plant metabolic status and PSA functioning. 

Overall, the variability of external stimuli and plant adaptation capabilities was sufficiently large to ascertain that the individual measurements of CF-based parameters was unsuitable for assessing the dormancy depth. Indeed, the same value of any given parameter could bear vastly different implications depending on weather patterns. However, time series collection and analysis have the potential to reveal the seasonal long-frequency response over the background of stochastic acclimatory changes occurring on the timescale of days and hours. The findings of this study demonstrated that both temperature and light intensity influence the PSA, inducing the photoprotective responses such as thermal dissipation of the absorbed light energy apparent as the measured non-photochemical quenching. The difference between the stimuli is that the sunlight follows strict rhythmicity, whereas the temperature changes in a more stochastic manner. Nevertheless, it was difficult to disentangle the response of the PSA to low temperature and high sunlight.

We hypothesized that the plant in the state of endodormancy exhibits less pronounced photoprotective response, likely because the dormant and hence metabolically quiescent plants do not need it. In this work, we made a step towards extracting useful information on winter dormancy from detailed time series of JIP-test parameters. Few research works dealing with CF in plants deal with time-domain analysis of chlorophyll fluorescence dynamics (for a remarkable example, see [68]), and those that do stop short of monitoring long-term processes such as winter dormancy. To the best of our knowledge, this work is the first demonstration of employing a combination of automated data collection across seasons and classic signal processing techniques to achieve data-driven noninvasive assessment of dormancy depth as well as contributions of individual processes affecting PSA during a winter period. 

Still, there are limitations to overcome. For example, while the suggested approach proved to be suitable for monitoring of dormancy status of plants, prediction of the timing of the onset and release of dormancy would require a more extensive observation across different plant species and climates.

To conclude, our study suggests the use of PAM fluorometry of chlorophyll as a powerful method for probing dormancy phases and stress resilience in plants. Its advantages include scalability and the possibility of non-invasive, automated measurements by many remote sensors connected to a computing cloud-based service for collection, analysis and visualization of the data. We believe that the developed approach has the potential to predict the aberrations of dormancy release in tree crops. Given this capability, fruit growers would be able to make timely informed decisions, e.g., on chemical interruption of dormancy to ensure normal blooming and fruiting. As an added benefit, this system could reveal severe frost damage, which is hard to diagnose visually before the beginning of the warm growing season. Implementation of such a system at scale would provide researchers with a tool with unprecedented time resolution and coverage for monitoring dormancy in wild and anthropogenic ecosystems. 

For the sake of robustness and to enable the use of time-frequency analysis, it is important to keep a continuous CF record. It is essential for suppressing momentary fluctuations caused by external factors such as weather conditions and other environmental stimuli. Defining thresholds for detecting the onset or release of dormancy will require additional research. The relevance and efficiency of the developed approach is supported by both destructive measurements and its good agreement with “traditional” chilling requirement models. Moreover, new such models could be developed with relative ease given that a sufficiently extensive CF record is available. Existing models are comparatively coarse and could be further fine-tuned, both for research and practical applications. Obviously, it would not be possible to pinpoint the exact moment of transition between endo- and eco-dormancy since it is unlikely that this transition is momentary. A more confident interpretation of the results of CF-based non-invasive probing of winter dormancy in deciduous plants will require a deeper understanding of the relationship between the phase of their dormancy and the functional state of their PSA.

## Figures and Tables

**Figure 1 plants-11-02811-f001:**
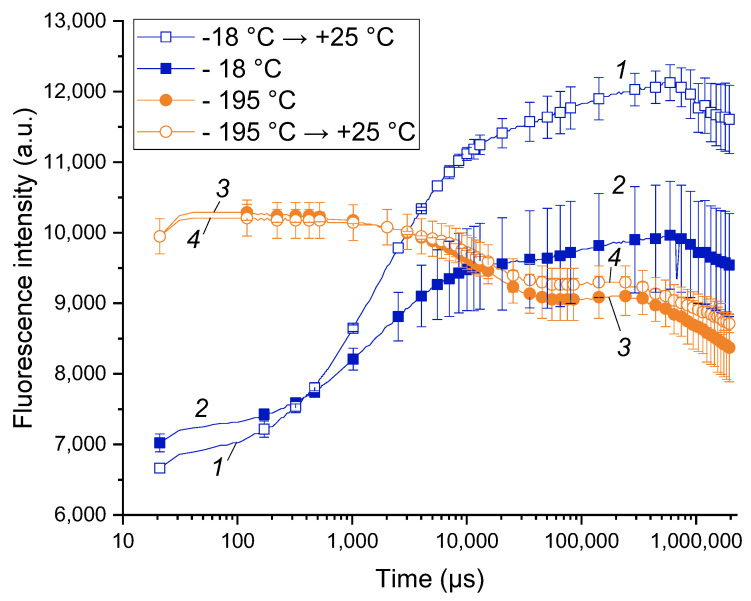
Difference between the chlorophyll a fluorescence induction curves of live apple tree shoots (1, 2) and those irreversibly damaged by liquid nitrogen freezing (3, 4). Typical OJIP curves of the apple tree shoots maintained at −18 °C (2) and subsequently warmed up to 25 °C (1) as well the OJIP curves of the apple tree shoots frozen in liquid nitrogen (−195 °C; 3) and subsequently warmed up to 25 °C (4).

**Figure 2 plants-11-02811-f002:**
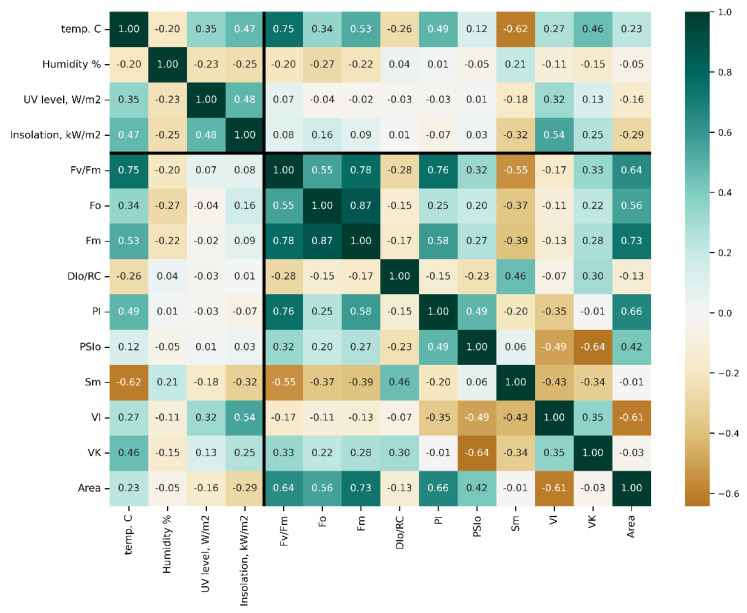
Pearson’s *r*-value matrix computed for pairs of the parameters studied in this work. Thick lines delineate fluorometry from meteorological data. For more detail on JIP-test parameters, see also [48] Appendix A.

**Figure 3 plants-11-02811-f003:**
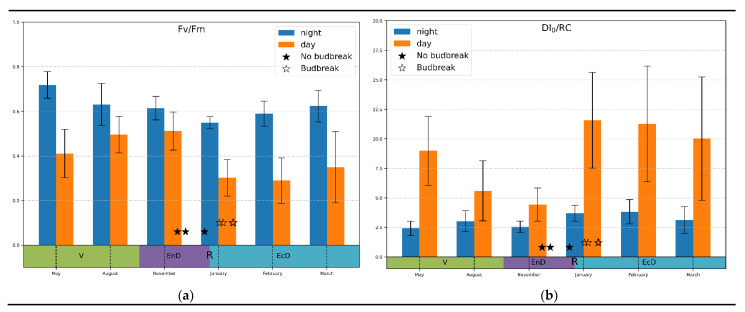
Monthly average values of (**a**) photosystem II quantum yield; (**b**) thermal dissipation per reaction center calculated at midnight and midday values (indicated on the graphs) during active vegetation (May), onset (August–November), maintenance (January) and release (February–March) of endo-dormancy. The annotations at the bottom denote approximate duration of dormancy and vegetation phases (V, vegetation period; EnD, endodormancy; EcD, ecodormancy; R, endodormancy release).

**Figure 4 plants-11-02811-f004:**
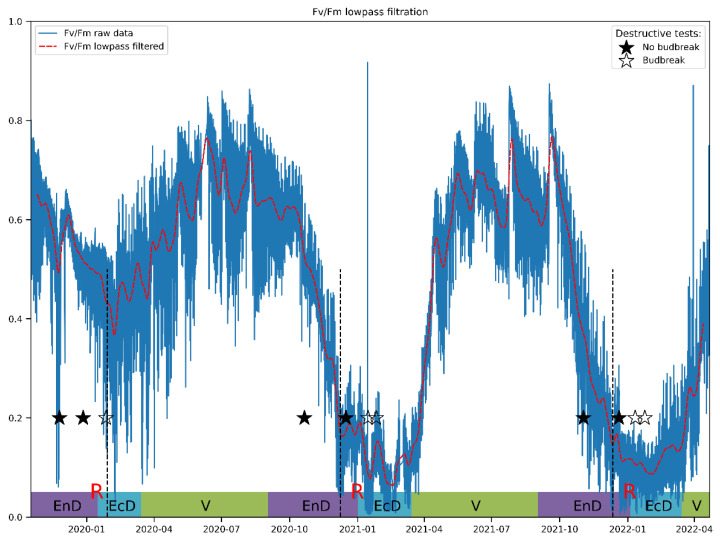
A typical view of an actual Fv/Fm record and its general trend obtained by the lowpass filtering of the respective time series. The annotations at the bottom denote approximate duration of dormancy and vegetation phases (V, vegetation period; EnD, endodormancy; EcD, ecodormancy; R, endodormancy release; destructive tests: ★, no budbreak displayed by the cut shoots; ☆, budbreak in room conditions). Vertical dashed lines denote the date of endodormancy release according to our model.

**Figure 5 plants-11-02811-f005:**
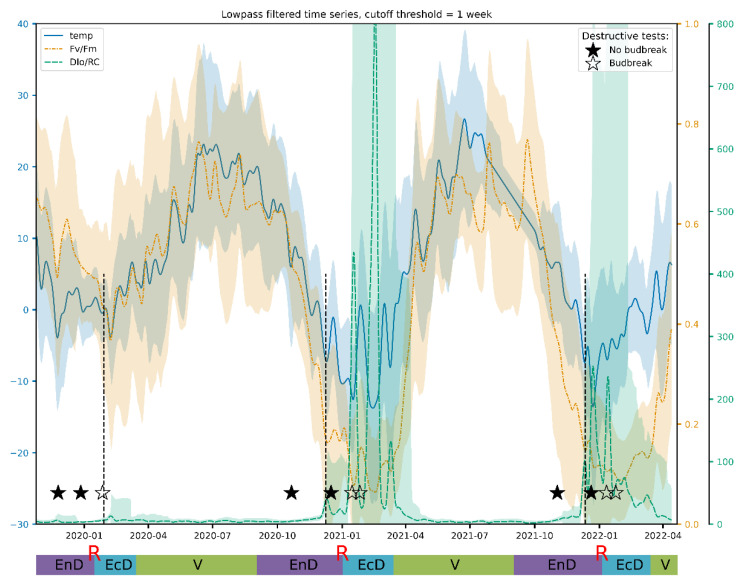
The time-courses of air temperature and the spectral power of the PS II Qy (Fv/Fm) and thermally dissipated energy flux (DI_0_/RC) variations with the period of one week (lowpass filtered value ± 2 × STD). The plots and the corresponding *Y*-axis are drawn in same color. For meaning of the annotations, see the legend to Figure 4.

**Figure 6 plants-11-02811-f006:**
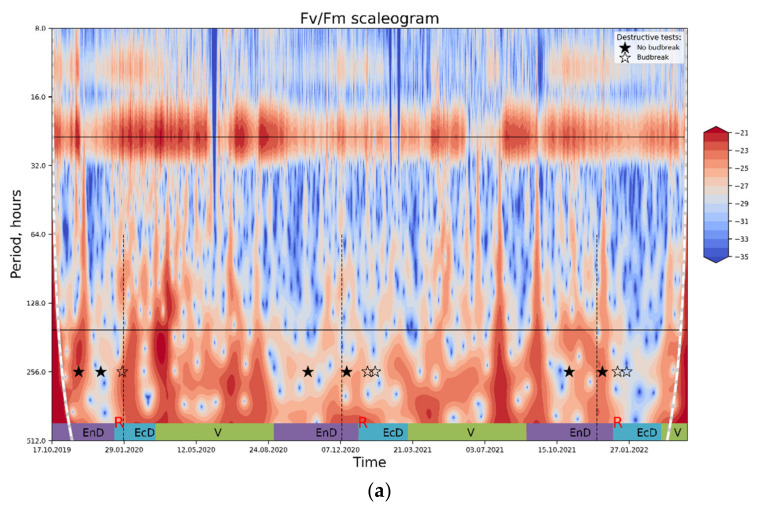
The results of wavelet transform analysis of (**a**) PS II Qy; (**b**) thermally dissipated energy flux per reaction center. An increase in spectral power pertinent to the processes influencing the OJIP parameter in question (red-colored regions on the plot) is tentatively ascribed to the periods of end-dormancy release and the onset of eco-dormancy. For explanations, see text. Annotations: top horizontal line, a period of 24 h; bottom horizontal line, a period of one week; rest-see the legend to Figure 4.

**Figure 7 plants-11-02811-f007:**
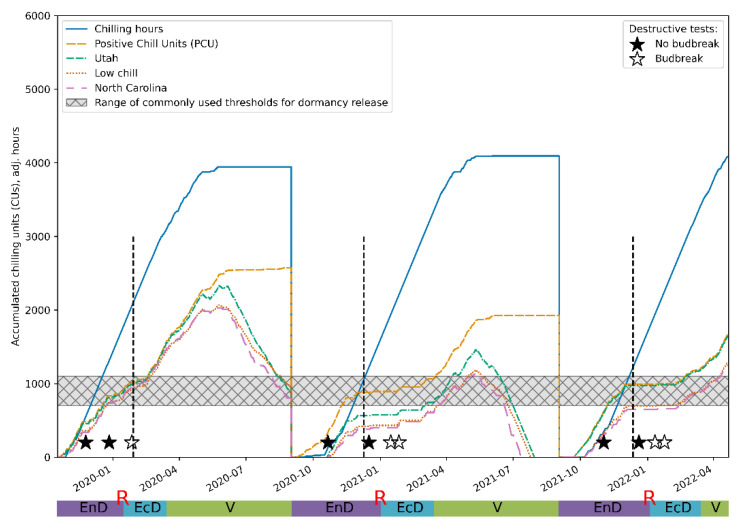
Comparison of the predictions of the endo-dormancy obtained using the wavelet analysis of the DI_0_/RC time series (see Figure 5 and Figure 6) and those inferred from the established CR models (see Materials and Methods). The vertical dashed lines denote the transition from endo-dormancy to eco-dormancy.

**Figure 8 plants-11-02811-f008:**
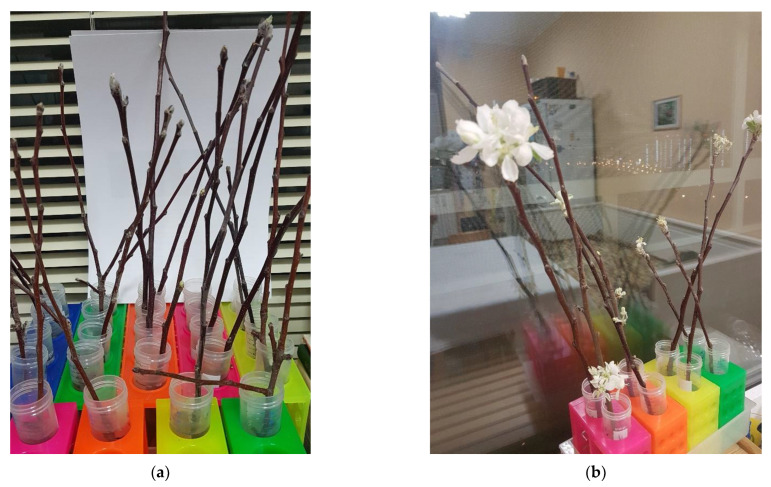
Release of endodormancy: (**a**) budbreak and (**b**) flowering of the apple tree shoots collected in February 2021 and grown in tap water at room temperature.

**Figure 9 plants-11-02811-f009:**
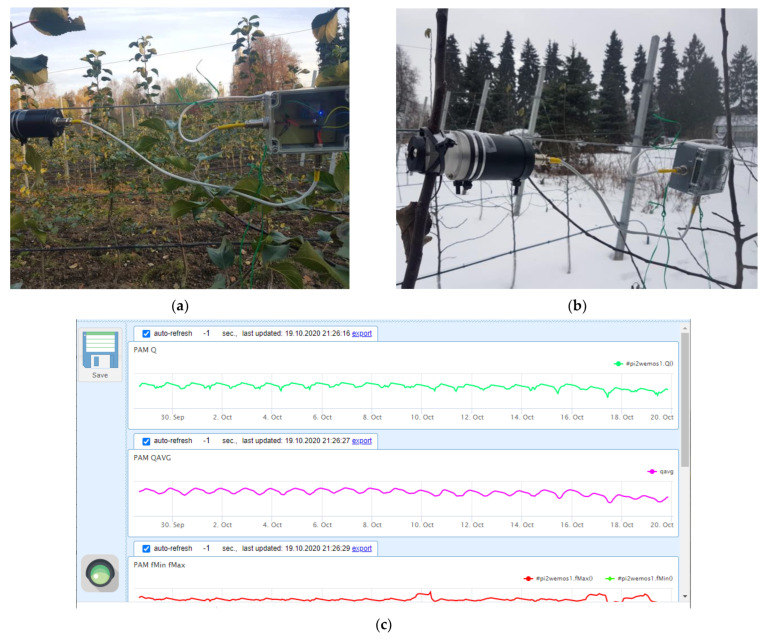
Sensors of variable chlorophyll fluorescence mounted on the monitored trees in the experimental orchard of Lomonosov Moscow State University in (**a**) fall and (**b**) winter seasons along with (**c**) a representative screenshot with hourly logged values of the OJIP parameters.

**Table 1 plants-11-02811-t001:** Results of the destructive tests of the release of endodormancy conducted in parallel with non-invasive assessments.

Sampling Time	% of Shoots Displaying Budbreak ^1^		
	2019	2020	2021
November	0	0	0
December	0	10±5	7 ± 4
January	65 ± 15 ^2^	85 ± 6	92 ± 6
February	75 ± 4	96 ± 3	98 ± 3

^1^ Within one week of incubation at room temperature, see Methods. ^2^ Average ± standard deviation values for five trees (see Appendix A) are shown.

## Data Availability

The raw data and derived parameters are available from the corresponding author on reasonable request. Code used in the analysis, accompanied with a subset of the data, is available on GitHub (https://github.com/Lodinn/PAM-timeseries, accessed on 20 September 2022).

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
