# Peer review of "Non-Invasive Probing of Winter Dormancy via Time-Frequency Analysis of Induced Chlorophyll Fluorescence in Deciduous Plants as Exemplified by Apple (*Malus* × *domestica* Borkh.)"

_plants, 2022, doi:10.3390/plants11212811_

Round 1

Reviewer 1 Report

Comments to authors

The purpose of this work was to test and validate a non-invasive method for dormancy monitoring in field-grown perennial trees.  The authors selected domestic apple as their species of interest, and carried out monitoring on 1 (then 2) trees over 3 winters starting in 2020.  The monitoring was compared with a traditional destructive method (collecting and indoor flushing of branch samples).  The authors found that their sample monitoring was successful across the seasons, with winter of 2020 being unusually warm.  The following two cold seasons were more typical, which allowed for better validation.  This paper presents an interesting method that seems highly applicable to orchards, wild forests, and more.  The remote nature of the sensing and non-destructive nature of the test makes this type of dormancy monitoring useful in real-world situations.  Overall, this was an interesting read.  A few additions and alterations will help strengthen the manuscript.  

Major

11.       The methods section needs some additional information.  In particular, how many trees were included in the field site?  Were samples collected from the same trees at each timepoint, or from different trees (the trees were small at the start of the study so it would be typical to collect from different individuals to avoid too much canopy loss).  Were samples collected from interior or border trees?  A field map showing the locations of the trees being monitored and the trees being sampled would be helpful.  Some of the parameters shown in figure 2 need to explained/defined such as area. 

22.       The introduction could be expanded a bit to include a mention of deciduous trees of economic importance (such as apple, the species of interest, other possible species are cherry, plum, peach etc.).  Please include the CR requirement of apple as well. 

33.       Please present quantified branch flush data for all the timepoints and years. 

Minor

44.       A helpful addition would be a graph of low temperatures collected across the seasons.  It would help to show the difference between the unusually warm 2020 winter and the following more typical cold seasons. 

55.       A quick discussion of why temperature was more influential on tree PSA than light intensity would be helpful.  After all, if the temperature is low enough there is no liquid water available for biological reactions, regardless of sun intensity. 

66.       The data in Figure 8 are mentioned first, please renumber the figures to match when they are first mentioned in the text. 

77.       Please avoid using red/green as informative colors (such as in Figure 1), not all readers can distinguish between these colors. 

88.       There are a few minor typos in need of correction.  Please italicize genus/species names (such as on line 86, line 249).  Define OJIP at first usage (line 141).  There is an extra comma at the start of line 149. 

99.       To clarify, were the -195 branches shown in Figure 1 measured while held at -195? 

110.   What are the vertical dashed lines in Figure 2?

111.   In figure 3, the V EnD EcC labels overlap the y-axix.  Please move them (or make them semi-transparent) to avoid blocking the text.  If there are data for when field bud break happened please add these data to the figure as well. 

112.   If there are field flush data for each year, please add them to figure 5.

113.   Figure 9 has images of the indoor bud flush for panels a and b instead of images of the field sensors, please switch to the correct pictures. 

Author Response

The purpose of this work was to test and validate a non-invasive method for dormancy monitoring in field-grown perennial trees.  The authors selected domestic apple as their species of interest, and carried out monitoring on 1 (then 2) trees over 3 winters starting in 2020.  The monitoring was compared with a traditional destructive method (collecting and indoor flushing of branch samples).  The authors found that their sample monitoring was successful across the seasons, with winter of 2020 being unusually warm.  The following two cold seasons were more typical, which allowed for better validation.  This paper presents an interesting method that seems highly applicable to orchards, wild forests, and more.  The remote nature of the sensing and non-destructive nature of the test makes this type of dormancy monitoring useful in real-world situations.  Overall, this was an interesting read.  A few additions and alterations will help strengthen the manuscript.  

RESPONSE:  we appreciate the careful analysis and positive evaluation of our manuscript by the reviewer. Please see below our detailed responses to the comments made/questions raised by the reviewer. Specific amendments made to the manuscript are highlighted using Tracking Changes mode in the manuscript itself (attached to the submission along with the “clean” manuscript file).

The methods section needs some additional information.  In particular, how many trees were included in the field site? 

RESPONSE:  the field site (0.1 ha) included 400 apple trees in total, 4000 plants/ha planting density. Now we mention this in the Materials and Methods (section 3.1).

Were samples collected from the same trees at each timepoint, or from different trees (the trees were small at the start of the study so it would be typical to collect from different individuals to avoid too much canopy loss)

RESPONSE:  the shoots for indoor incubation experiments were taken from five trees standing in row. These five trees included (i) the two trees from which the Chl fluorescence data were recorded and (ii) one tree between those trees and (iii) the two flanking trees. Please also see new Fig. S1.

Were samples collected from interior or border trees?  

RESPONSE:    No, the samples and the data were taken from the trees shielded by another trees; please see new Fig. S1.

A field map showing the locations of the trees being monitored and the trees being sampled would be helpful. 

RESPONSE:    The corresponding scheme has been added; please see new Fig. S1.

Some of the parameters shown in figure 2 need to explained/defined such as area. 

RESPONSE:    The explanation has been added to Table S1.

The introduction could be expanded a bit to include a mention of deciduous trees of economic importance (such as apple, the species of interest, other possible species are cherry, plum, peach etc.).  Please include the CR requirement of apple as well.

RESPONSE:    The introduction was expanded with appropriate references highlighting observed and projected climate change related-issues in other commercially significant cultivars. Role of apple as one of the tree crops most vulnerable to insufficient chill was highlighted and reference values for CR were provided (see the text of the revised version of the manuscript).

Please present quantified branch flush data for all the timepoints and years.

 RESPONSE:   The required date has been provided as Table 1.

Minor

A helpful addition would be a graph of low temperatures collected across the seasons.  It would help to show the difference between the unusually warm 2020 winter and the following more typical cold seasons. 

RESPONSE:    a graph of temperatures collected across the seasons could be seen as a part of Figure 5 (solid blue line). It is apparent that the 2020 winter was significantly warmer than the two following ones, which were more typical for the region; in 2019/2020 low temperatures were hardly ever reaching the –10 °C mark.

A quick discussion of why temperature was more influential on tree PSA than light intensity would be helpful.  After all, if the temperature is low enough there is no liquid water available for biological reactions, regardless of sun intensity. 

RESPONSE:    actually, our point is that both temperature and light intensity do influence the PSA inducing the photoprotective responses including thermal dissipation of the absorbed light energy apparent as the measured non-photochemical quenching. The difference between the stimuli is that the sunlight follows the strict rhythmicity whereas the temperature changes in a more stochastic manner. Nevertheless, it is difficult to disentangle the response of the PSA to low temperature and high sunlight. Essentially, we hypothesized that the plant in the state of endodormancy exhibits less pronounced photoprotective response (likely because plant do not need it due to metabolic quiescence under endodormancy). We demonstrated that this effect can be pinpointed by the time series analysis of Chl fluorescence parameters. We amended the Concluding remarks to make this point more clear.

The data in Figure 8 are mentioned first, please renumber the figures to match when they are first mentioned in the text.

RESPONSE:    The reference to Figure 8 at the beginning of the Results section was deemed                               unnecessary and was removed. Having this done, all the other figures were numbered                 in proper order.

Please avoid using red/green as informative colors (such as in Figure 1), not all readers can distinguish between these colors.

RESPONSE:    colors on the offending figure were adjusted accordingly. Additionally, all of the figures in the manuscript were passed through the color vision deficiency simulation software and two more figures—Fig. 5 and Fig. 7—also underwent minor corrections as a result.

There are a few minor typos in need of correction.  Please italicize genus/species names (such as on line 86, line 249).  Define OJIP at first usage (line 141).  There is an extra comma at the start of line 149. 

RESPONSE: the typos were corrected, thank you.

To clarify, were the -195 branches shown in Figure 1 measured while held at -195? 

RESPONSE:    no, they were kept for 30 min at –195 and measured after thawing at room temperature. The clarification has been added to the text.

What are the vertical dashed lines in Figure 2?

RESPONSE:    There are no vertical dashed lines in Figure 2. Label to Figure 4 was faulty, however - thank you for bringing this to our attention. All the figure legends were reviewed and those concerning annotations were streamlined to avoid excessive (to the fault) cross-referencing.

In figure 3, the V EnD EcC labels overlap the y-axix.  Please move them (or make them semi-transparent) to avoid blocking the text.  If there are data for when field bud break happened please add these data to the figure as well. 

RESPONSE:    There was, indeed, a formatting issue. Both panels of figure 3 were rebuilt to prevent this from happening again.

If there are field flush data for each year, please add them to figure 5.

RESPONSE:    We mentioned in the text of the revised version that in the field, the trees flushed in the third decade of April.

Figure 9 has images of the indoor bud flush for panels a and b instead of images of the field sensors, please switch to the correct pictures. 

RESPONSE: Thank you; the images were replaced.

Reviewer 2 Report

This is a very interesting project, and the results are promising.

I have some comments and questions.

Line 346 in M&M, how many is “several times” and what are the room conditions?

In my opinion, the results of the regression analysis must be improved by providing at least tests of the parameters (p-values), and RSME. Instead of color in the correlation matrix is better to include the significance tests or p-values.

Figures 3a and 3b should be fixed.

There is confusion between variables and parameters, for example, weather parameters instead weather variables.

You mention the annotations of figure 4 can be seen in figure 2 but figure 2 does not have any annotation.

The title on the top of some graphs is not necessary.

What is the meaning of CU in figure 7?

It is indispensable to include a table providing the convections used for the variables.

Author Response

(The authors gave the same response as above.)

Reviewer 3 Report

Excellent work.

Author Response

RESPONSE:    we are deeply grateful for such a kind assessment of our work.

Reviewer 4 Report

Comments to the Authors

I have reviewed with interest your manuscript entitled „Non-invasive probing of winter dormancy via time-frequency 2 analysis of induced chlorophyll fluorescence in deciduous 3 plants as exemplified by apple (Malus × domestica Borkh.)” submitted to Plants.

In my opinion the current version of your manuscript is suitable for publication in Plants, but after same small correction. The quality of the presentation could be improved-e.g. In general, manuscript needs only small improvement.

The article suffers from a number of small mistakes, ranging from misspellings to incorrectly phrased sentences.

Some adjustments are suggested to qualify the paper:

Issues include:

The Abstract

The abstract should be a total of about 200 words maximum. Now it exceed 200 words (now there is 241words). I propose to shorten it.

The general comment to the Introduction section: the introduction is written in an appropriate manner. The content of the literature review chapter is related to the research topic. Up-to-date literature references are presented in the manuscript by the author, but there are same references before 2010 – 26%. I suggest to put more attention and write the main and detailed aim of the study at the end of the Introduction.

In the chapter Materials and Methods, the methodology is adequate, but there is a lack of information in some aspects. Some more information should be explained in the text. How many repetition have been used in the experiment? Could Author(s) of manuscript add more detailed information of methods of statistical analysis? Could the Author(s) explain it? Moreover, why the subsection entitled: Materials and Methods have been put after subsection Results and discussion.

In the chapter Results, the results are displayed correctly.

The Discussion is informative. Moreover, the Authors attempt to discuss their important results and the rest is a quotation of literature.

In my opinion the subsection  entitled Concluding remarks is insufficiency. It could be contain more obtained by Authors results and same recommendations for farmers, advisers and other recipient of study results. The subsection have to be more essential and clear.

More small recommendation there are in the text of manuscript.

I hope that these comments help you to make an improved the final version of the manuscript.

Author Response

I have reviewed with interest your manuscript entitled „Non-invasive probing of winter dormancy via time-frequency 2 analysis of induced chlorophyll fluorescence in deciduous 3 plants as exemplified by apple (Malus × domestica Borkh.)” submitted to Plants.

In my opinion the current version of your manuscript is suitable for publication in Plants, but after same small correction. The quality of the presentation could be improved-e.g. In general, manuscript needs only small improvement.

The article suffers from a number of small mistakes, ranging from misspellings to incorrectly phrased sentences

RESPONSE:  we appreciate the careful analysis and positive evaluation of our manuscript by the reviewer. Please see below our detailed responses to the comments made/questions raised by the reviewer. Specific amendments made to the manuscript are highlighted using Tracking Changes mode in the manuscript itself (attached to the submission along with the “clean” manuscript file).

The abstract should be a total of about 200 words maximum. Now it exceed 200 words (now there is 241words). I propose to shorten it.

RESPONSE:    we did our best to shorten the abstract without losing essential information.

The general comment to the Introduction section: the introduction is written in an appropriate manner. The content of the literature review chapter is related to the research topic. Up-to-date literature references are presented in the manuscript by the author, but there are same references before 2010 – 26%. I suggest to put more attention and write the main and detailed aim of the study at the end of the Introduction.

RESPONSE:    we introduced some newer literature sources to the reference list (new refs. 16–21).

In the chapter Materials and Methods, the methodology is adequate, but there is a lack of information in some aspects. Some more information should be explained in the text. How many repetition have been used in the experiment?

RESPONSE:    this information is now presented in the text. Briefly, we have hourly continuous record of the fluorescence parameters from two trees for three years in row. As a ground truth, we have destructive tests involving the shoots from five trees (see the new Table 1).

Could Author(s) of manuscript add more detailed information of methods of statistical analysis?

RESPONSE:    as we mentioned above, we are handling large time series and there are thousands of measurements for each parameter. Owing to that, p-values for all pairwise correlations that are not very close to zero are tiny. For the parameters discussed, most p-values are on the order of 1e-200, with some falling under the standard available precision and coming out of the analysis toolbox as just zeroes. A short statement reflecting that was added to the beginning of Section 2.3. Please see also supplementary methods: The same logic about the sample sizes applies to the regressions provided in part in Supplementary Materials (p-values about 1e-193 and 1e-243). The RMSE values for those regressions were added to the text.

Could the Author(s) explain it?

RESPONSE:    please see the Theoretical background section and other sections in the Supplementary material.

Moreover, why the subsection entitled: Materials and Methods have been put after subsection Results and discussion.

RESPONSE:    this was done according to the requirements of the Plants journal standards.

In my opinion the subsection  entitled Concluding remarks is insufficiency. It could be contain more obtained by Authors results and same recommendations for farmers, advisers and other recipient of study results. The subsection have to be more essential and clear.

RESPONSE:    We believe that the developed approach has a potential to predict the aberrations of dormancy release in tree crops. Given this capability, fruit growers would be able to make timely informed decisions e.g. on chemical interruption of dormancy to ensure normal blooming and fruiting. We added this and some other passages to the revised text of the mansucript.

More small recommendation there are in the text of manuscript.

I hope that these comments help you to make an improved the final version of the manuscript.

RESPONSE:    thank you very much, these recommendations have been taken into account in the revised version of the manuscript.